# Variational Constrained Reinforcement Learning with Application to Planning at Roundabout

## Abstract

Planning at roundabout is crucial for autonomous driving in urban and rural environments. Reinforcement learning is promising not only in dealing with complicated environment but also taking safety constraints into account as a as a constrained Markov Decision Process. However, the safety constraints should be explicitly mathematically formulated while this is challenging for planning at roundabout due to unpredicted dynamic behaviour of the obstacles. Therefore, to discriminate the obstacles' states as either safe or unsafe is desired which is known as situation awareness modeling. In this paper, we combine variational learning and constrained reinforcement learning to simultaneously learn a Conditional Representation Model (CRM) to encode the states into safe and unsafe distributions respectively as well as to learn the corresponding safe policy. Our approach is evaluated in using Simulation of Urban Mobility (SUMO) traffic simulator and it can generalize to various traffic flows. (Anonymous code is available to reproduce the experimental results and additional videos are also available [1].)

## 1 Introduction

The recent progress in model-free reinforcement learning (RL) (Sutton et al., 1992) has produced many interesting results in planning and control problems and proves to be effective in finding optimal policy for nonlinear stochastic systems when the dynamics are either unknown or affected by severe uncertainty (Buşoniu et al., 2018), including complicated robotic locomotion and manipulation (Kumar et al., 2016; Xie et al., 2019; Hwangbo et al., 2019). To further ensure the safety in control, applying constraints over states and actions is a natural way. Typically, a standard and well-studied formulation for reinforcement learning with constraints are the constrained Markov Decision Process (CMDP) framework (Altman, 1999), where the discounted sum of safety cost should be under certain bounds. As a consequence, discount parameter or bound has to be tuned to ensure safety (García & Fernández, 2015).

However, in many complicated scenarios when the traffic flow changes frequently, the mathematical description of the obstacles are difficult to obtain explicitly. In this scenario, the constraints are dynamic with uncertainty, which is challenging to define proper constraint. Besides, it is still challenging to tune a cost function with such complicated constraints. Without certain predefined constraints, existed methods fail to find a feasible policy.

When humans driving a car, they do not know any specific safety constraint and still perform well. For example, at the very beginning, one is always not sure about the distance between the headstock and wall or other vehicles, which may lead to some accidents. However, with more driving experience, the better awareness of safety and latent constraints will be slowly formed. In this scenario, the constraint is not formulated as any specific distance between obstacles but the feeling according to experience. One of the talents of humans is that humans heavily rely on to make safe decisions is that we could understand the latent information of complicated situation or environment and awareness, whether it is safe or dangerous (Bubic et al., 2010). This capability is called situation awareness (SA) (Endsley, 2017). In RL, SA could be regarded as a model that can accurately aware current situation,

---

[1] https://www.dropbox.com/sh/oo6zty99c6tclx1/AAA8RXynrE8K9SYpxzqBhv4Va?dl=0

internally represent the complex dynamics and covering enough latent information like constraints. Furthermore, it should be able be embedded into RL framework to guide the agent how to safely interact with environment.

This paper proposes a novel approach for encoding the measure of safety in scenarios where the explicit safety cost is not available, or the states are interfered by severe uncertainties. The contribution of this paper is in two-folds: 1) present a variational-based method to encode the safe and unsafe states; 2) measure the level of safety in latent space with Wasserstein distance, taking the uncertainty in states into consideration. Our framework can be generally combined with various RL algorithms, improving the performance of RL algorithms in terms of safety. Finally, we evaluate our approach in roundabout task with different traffic flow and show that our approach significantly improves the success rate of baseline.

## 2 RELATED WORK

There are several prior works about the reinforcement learning with safety constraints have been done. Achiam et al. (2017) proposed a safety constrained policy optimization (CPO) approach based on the trust region method, which guarantees the constraint satisfaction with a safe initial policy. Wen & Topcu (2018) came up with a constrained cross-entropy method for finite CMDP tasks, which an effectively learn feasible policies with respect to constraint satisfaction. Chow et al. (2019) came up with Lyapunov-based approach and combine with both on-policy and off-policy reinforcement learning algorithms, which achieves better results in terms of balancing the performance and constraint satisfaction compare with CPO. However, both of the results above need full knowledge of the constraints, and the tasks are almost static environment without uncertainty, rather than a complex dynamic case with many unpredictable obstacles, which is the focus of this paper.

Situation awareness is the key to successful decision-making (Nullmeyer et al., 2005). Recently, many works on reinforcement learning with situation awareness have been done. Teng & Tan (2008) proposed TD-FALCON methods that could integrate Context-aware Decision Support (CaDS) system and learning for supporting context-aware decision making, where CaDS system exploits contextual information for focused situation assessment and goal-oriented decision support. D'Aniello et al. (2014) proposed Context Space Theory (CST) to represent raw data in high-level, domain-relevant concept, namely context attribute, which could identify situations the user is involved in and considering user's situated preferences. Yin et al. (2019) propose two new metrics $CTS_a$, $CTS_n$ to assess the taxi situation, and they found a significant relationship is revealed between the taxi delay and $CTS_a$ at Level-1, and the taxi time and $CTS_n$ at Level-2, which provides strong reference to airport ground movements for control and management purposes. However, all of which are focused on a simple task or static environment. Most importantly, the proposed situation awareness models are all deterministic, which might lose much information in the task with the action-conditioned settings and dynamics uncertainty.

## 3 PROBLEM FORMULATION

Reinforcement learning with safety constraints tends to be a promising way to offer intelligence and safety simultaneously for autonomous driving. In this paper, we will focus on the decision and control tasks with safety constraints which can be modeled by constrained Markov decision process (CMDP) (Altman, 1999). A CMDP is a tuple, $(\mathcal{X}, \mathcal{A}, r, P, c, \rho)$, where $\mathcal{X}$ is the set of states and $\mathcal{A}$ is the set of actions. $P(x'|x, a)$ is the transition probability function and $\rho(x)$ is the starting state distribution. $r(x, a)$ and $c(x, a)$ are the reward the constraint function, respectively.

In CMDP where the constraints $c$ is not well defined, we aim to simultaneously find a policy $\pi$ and safety cost $\hat{c}$ such that

$$\max_{\pi} \mathbb{E}_{\tau \sim \pi}[\sum_{t=0}^{\infty} \gamma^t r(x_t, a_t)] \tag{1}$$

$$\text{s.t. } \mathbb{E}_{\tau \sim \pi}[\sum_{t=0}^{\infty} \gamma^t \hat{c}(x_t, a_t)] \leq \hat{d} \tag{2}$$

where $\gamma \in [0, 1)$ is the discount factor, $\tau$ denotes a trajectory ($\tau = (x_0, a_0, s_1, ...)$), and $\tau \sim \pi$ is shorthand for indicating that the distribution over trajectories depends on $\pi$: $x_0 \sim \rho$, $a_t \sim \pi(\cdot|x_t)$, $s_{x+1} \sim P(\cdot|x_t, a_t)$. $\hat{d}$ is the learned safety threshold. Though the form of $c$ is unknown, we assume that the data set composed of states labeled as safe or dangerous is available, i.e. $\{(x_1, s_1), (x_2, s_2), (x_3, s_3), ... \}$ where $s \in \{0, 1\}$ indicates whether the state is safe or dangerous. Note that it is possible the same state $x$ has different labels at different data points, since various actions may be taken. Based on this data set, we aim to learn a model $q_\phi(z|x)$ to map the states to latent space and then construct a safety cost $\hat{c}$ based on this latent model.

## 4 MAIN RESULTS

Inference the implicit constraints and awareness of a state to be safe or dangerous are the essential capability for safe decision making. To model the awareness of constraints, the key is to map from physical state space to a latent space where represents the implicit constraints. We proposed a novel method could infer the safety condition as well as the latent constraints.

In this section, our approach for inference of the latent constraints will be described in detail. First, in Section 4.1, a conditional representation model is proposed to map the states into the latent space, where the latent space of safe and dangerous states are separated from each other. Then in Section 4.2, we employ Wasserstein distance in measuring level of safety of the given state in latent space, i.e. the constructed safety cost $\hat{c}$.

### 4.1 CONDITIONAL REPRESENTATION MODEL

In this part, we propose the conditional representation model (CRM) that is capable of mapping the state space to a latent variable space $\mathcal{Z}$. For this purpose, we borrow the encoder-decoder structure from variational autoencoder (VAE) (Kingma & Welling, 2013; Doersch, 2016), which provides an efficient approach for approximating posterior distribution of the latent variable $z$ given an observation $x$. For normal VAE, the variational bound is written as

$$\log p_\theta(x) = D_{\mathrm{KL}}(q_\phi(z|x)\|p_\theta(z|x)) + L(x; \theta, \phi) \tag{3}$$

$$L(x; \theta, \phi) = -D_{\mathrm{KL}}(q_\phi(z|x)\|p_\theta(z)) + \mathbb{E}_{q_\phi}[\log p_\theta(x|z)] \tag{4}$$

where $q_\phi(z|x)$ is the recognition model, an approximation to the intractable true posterior $p_\theta(z|x)$. $p_\theta(x|z)$ is the generative conditional distribution and $p_\theta(z)$ is an arbitrary prior distribution. $D_{KL}$ denotes the Kullback-Leibler (KL) divergence.

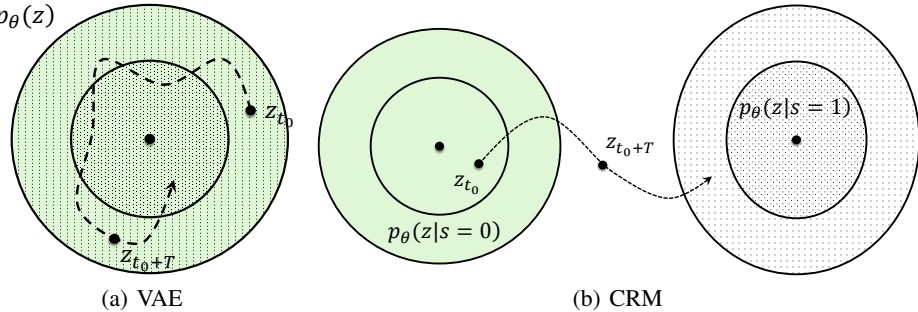

(a) VAE            (b) CRM

Figure 1: **a).** For **VAE**, the green circle denotes the prior distribution. All the latent distributions from both safe and dangerous states map into a compact region near each others, which has no capability to distinguish the safe and dangerous state**b).** For **CRM**, there are two latent space. We hope all the dangerous state maps to the white space after inference, while normal state maps to the green space after inference. The figure shows when the trajectories drive into a dangerous situation, the latent distribution will move close to white latent space from green latent space. Our method is able to separate the safe and dangerous latent space.

In our framework, the CRM is not only aimed at encoding, but also contains enough information for measuring how safe or unsafe the current latent is, which is not a concern in classical VAE. In

equation 3, the latent of both safe and dangerous states are sampled from the same prior distribution $p_\theta(z)$ (as shown in Figure 1(a)), resulting in that distributions of latent variable for safe and dangerous states are hardly separable, which brings trouble for measuring the safety in the following step.

The data set for CRM is composed of a series of data points containing both the state and its label, $\{(x_1, s_1), (x_2, s_2), (x_3, s_3), \dots\}$. To exploit the binary classification label $s$ in the process of latent representation learning, we rewrite the marginal likelihood in Eq.(3) as

$$\mathbb{E}_{p(s)} \log p_\theta(x|s) = \mathbb{E}_{p(s)}(D_{\mathrm{KL}}(q_\phi(z|x)\|p_\theta(z|x,s)) + L(x;\theta,\phi,s)) \tag{5}$$

$$L(x;\theta,\phi,s) = -D_{\mathrm{KL}}(q_\phi(z|x)\|p_\theta(z|s)) + \mathbb{E}_{q_\phi}[\log p_\theta(x|z,s)] \tag{6}$$

Here, the objective is to use an encoding model $q_\phi(z|x)$ to approximate the conditional posterior $p_\theta(z|x,s)$, which is conditioned on both the state and its label. This is equivalent to maximizing the lower bound in Eq.(6). Additionally, we would like the $z$ for safe and dangerous states are separately distributed as shown in Figure 1(b), so that we are able to measure the safety of a given state $x_t$ by calculating the divergence between $q_\phi(z_t|x_t)$ and the distribution of latent for dangerous states $p_\theta(z|s = 1)$. Take the trajectory in Figure 1(b) as example, as the divergence function $D(q(z_t|x_t), p_\theta(z|s = 1))$ decreases, it is intuitive that the state $x_t$ is becoming more and more unsafe. The discussion on how to exploit specific $D$ to construct safety cost $\hat{c}$ is deferred to the next subsection.

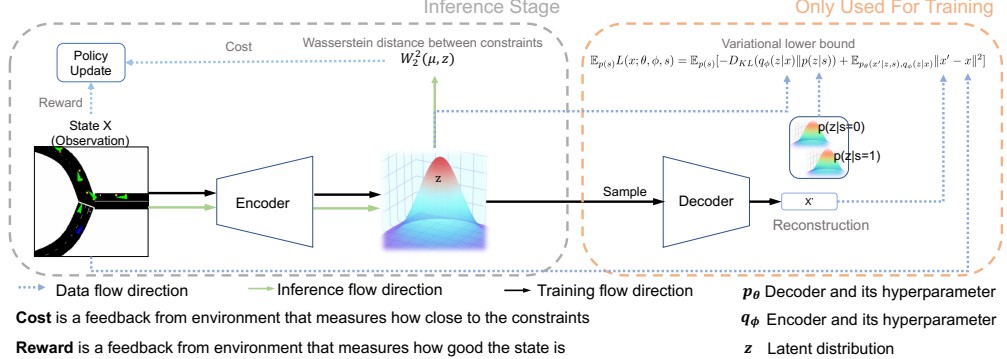

Figure 2: Overview of the data flow of Conditional Representation Model for both training and inference stages.**1). Training Stage**: During interaction with the environment, the agent collects both dangerous and safe states. Through the encoder $q_\phi$, where $\phi$ are the hyperparameters of the network (more details on the neural network architecture can be found in Appendix B),we expected all the dangerous state maps to the arbitrary prior distribution $p(z|s = 1)$, and all the safe state maps to the another arbitrary prior distribution $p(z|s = 0)$, contributing to the first term of the objective function $\min D_{\mathrm{KL}}(q_\phi(z|x)\|p_\theta(z|c))$. To verify the inferred distribution carrying effective information, we sample from inferred latent distribution $q_\phi(z|x)$ then use a decoder to reconstruct the original data based on the samples, forming the second term of the objective function $\|x' - x\|^2$. **2). Inference Stage**: The agent uses encoder to infer latent information and formulates safety cost by Wasserstein distance. Finally,an CMDP reinforcement learning algorithm is employed to update the policy with safety cost and reward.

Before proceeding, there are several issues to be addressed. First, different to the lower bound in VAE, the objective we aim to maximize is also conditioned on label $s$. If a task is highly safe or unsafe, i.e. $p(s = 1)$ or $p(s = 0) \approx 0$, then the objective falls back to the form in VAE, since the rare part of data plays an insignificant role in the approximation loss. Thus it is necessary to strike a balance between the amount of safe and dangerous data. Secondly, in this paper, we use Gaussian distributions with different expectations as the prior distributions, i.e. $p_\theta(z|s = i) = \mathcal{N}(\mu_{s=i}, I), i \in \{0, 1\}$. The expectations $\mu_{s=0}$ and $\mu_{s=1}$ are selected to be reasonably apart from each other to ensure that the safe and dangerous states are separable in the latent space.

In this paper, the encoder $q_\phi(z|x)$ is parameterized by a Gaussian MLP (a fully connected neural network) and decoder $p_\theta(x|z, s)$ parameterized by an MLP. For training of these networks, we come

up with the following empirical objective function,

$$\mathbb{E}_{p(s)}L(x; \theta, \phi, s) = \mathbb{E}_{p(s)}[-D_{KL}(q_\phi(z|x)\|p(z|s)) + \mathbb{E}_{p_\theta(x'|z,s),q_\phi(z|x)}\|x' - x\|^2] \quad (7)$$

where $x'$ is the generated state based on the latent sampled from $q_\phi(z|x)$. Since both the encoder network and the prior distributions are both Gaussian, the above equation could be further written as

$$\mathbb{E}_{p(s)}L(x; \theta, \phi, s) = \mathbb{E}_{p(s)}[(1 + \log(\sigma(x)^2) - (\mu(x) - \mu_s)^2 - \sigma(x)^2) \\ + \mathbb{E}_{p_\theta(x'|z,s),q_\phi(z|x)}\|x' - x\|^2] \quad (8)$$

where $\mu_s$ denotes the expectation of prior distributions. $\mu(x)$ and $\sigma(x)$ represents the expectation and diagonal elements of covaraiance matrix for the Gaussian MLP. The latent of a given state $x$ could be generated by

$$z = \mu(x) + \sigma(x) \odot \epsilon, \epsilon \sim \mathcal{N}(0, I) \quad (9)$$

To sum up, we expected all the dangerous state maps to the arbitrary prior distribution $p(z|s = 1)$, and all the safe state maps to the another arbitrary prior distribution $p(z|s = 0)$, contributing to the first term of the objective function $\min D_{\mathrm{KL}}(q_\phi(z|x)\|p_\theta(z|c))$. To verify the inferred distribution carrying effective information, we sample from inferred latent distribution $q_\phi(z|x)$ then use a decoder to reconstruct the original data based on the samples, forming the second term of the objective function $\|x' - x\|^2$.

Data flow between encoder and decoder and how each part is involved in the above objective function during training are shown in Figure 2.

## 4.2 Construction of Safety Constraints

In this part, the measurement of divergence between the approximated posterior latent distribution $q_\phi(z|x)$ and the prior $p_\theta(z|s = 1)$ is specified and exploited in constructing the safety cost $\hat{c}$ in equation 2.

First, for the measurement of divergence $D(q_\phi(z|x), p_\theta(z|s = 1))$, a number of common metrics exist, such as KL divergence (Hershey & Olsen, 2007) and Jensen-Shannon (JS) divergence (Lin, 1991). However, both KL and JS divergence are not suitable for measuring the divergence between distributions that are distinct from each other. Instead, we use Wasserstein distance to measure the divergence between the aforementioned distributions. Wasserstein distance(Panaretos & Zemel, 2019) is a metric of divergence inspired by the problem of optimal mass transportation. The $p$-Wasserstein distance between two distributions $p_1$ and $p_2$ on $\mathbb{R}^d$ is defined as:

$$W_p(p_1, p_2) = \inf_{X \sim p_1, Y \sim p_2} (\mathbb{E}\|X - Y\|^p)^{1/p} \quad (10)$$

where the infimum is taken over all pairs of $d$-dimensional random vectors $X$ and $Y$ marginally distributed as $p_1$ and $p_2$, respectively. Compared with other metrics, Wasserstein distance is more informative even when the distributions are far apart from each other, while KL-divergence goes to infinity and JS-divergence equals to a constant in this case. As $q_\phi(z|x)$ and $p_\theta(z|s = 1)$ are both Gaussian distributions, the Wasserstein distance between them could be analytically written as:

$$W_2^2(q_\phi(z|x), p(z|s = 1)) = \|\mu(x) - \mu_{s=1}\|^2 - \mathrm{tr}(I - \mathrm{diag}(\sigma(x))) \quad (11)$$

where the $\mathrm{tr}(\cdot)$ denotes the trace of a matrix and $\mathrm{diag}(\cdot)$ denotes forming a diagonal matrix with the given vector.

For CMDP tasks, the agent is required to keep away from the dangerous states (or constraints) beyond certain threshold. Thus, we propose a general Wasserstein-based cost as:

$$\hat{c}(x) = \max(\frac{\overline{d}}{W_2^2(q_\phi(z|x), p(z|s = 1))} - 1, 0) \quad (12)$$

where $\overline{d}$ is the threshold to be tuned. The safety cost decreases as Wasserstein distance increases and becomes zero when Wasserstein distance is larger than $\overline{d}$. Only if the the current state achieves the threshold of safety-level, there will be a cost, which means there will not any influence or restriction within the threshold of safety-level, as shown in Figure 3.

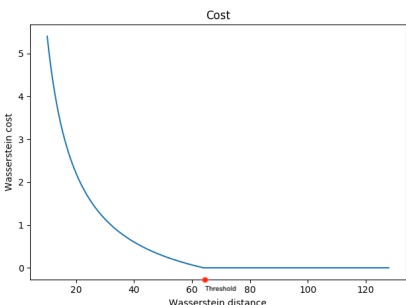

Figure 3: The curve of safety cost.X-axis indicates the value of Wasserstein distance and Y-axis indicates the value of safety cost. The red dot indicates the threshold of safe Wasserstein distance, under which the state is regarded as potentially dangerous.

---

**Algorithm 1** CRM-based Reinforcement learning

---

Randomly initialize neural networks of reinforcement learning algorithm, and encoder $q_\phi$, decoder $p_\theta$ of CRM
Assume dangerous prior distribution $\mu$, and safe prior distribution $\nu$
**for** each iteration **do**
    Sample $s_0$ according to $\rho$
    **for** each time step **do**
        Sample $a_t$ from policy $\pi$ and step forward
        Observe $s_{t+1}$ and $r_t$
        Evaluate $q_\phi(z_{t+1}|s_{t+1})$ with encoder and calculate $c_t$ according to (12)
        Store $(s_t, a_t, r_t, c_t, s_{t+1})$ for policy update
        **if** Dead or collision, store $s_{t-n:t+1}$ as dangerous state for CRM update
        **else** Store $s_{t+1}$ as safe state for CRM update
    **end for**
    **for** each update step **do**
        Update reinforcement learning policy
        Sample equal amount of data from safe states and dangerous states then update the CRM using gradient decent minimizing (8)
    **end for**
**end for**

---

Finally, we demonstrate how the proposed approach is incorporated with the general CMDP RL algorithms in Algorithm 1. During training, the data for training CRM is collected from the agent's interaction with the environment, while CRM provides the safety cost at the same time. Once the agent dies, the last $n$ steps state will be labeled as dangerous while others labeled as safe. For every update step, equal amount of data is sampled from the set of safe and dangerous states, then used for evaluating the gradient of encoder and decoder with respect to Eq.(8). If a datasets of safe and dangerous states are available before the training starts, it is also desirable to pre-train CRM in advance, which in general accelerates the convergence of policy learning.

## 5 EXPERIMENTS

We tested our algorithm on the task of driving through a double-lane roundabout with four exits. Figure 4 depicts a bird's-eye view of the roundabout.

The challenges faced in the roundabout include a large amount of interaction between the traffic participants, the complexity of other agents' driving behavior, the vast perceived uncertainty due to the geometry of the road and the continuous operation in a short period time (turn to the inner track, turn to outside lane, negotiation). Most importantly, it is difficult to define the constraints and design a cost function properly under dynamic traffic with uncertainty. Thus, we aimed to model the latent

safety constraints with CRM and employ the CMDP RL algorithm together with the learned safety cost to find a safe policy.

For the experimental environment, the roundabout scenario is simulated by the Simulation of Urban Mobility (SUMO) traffic simulator (Behrisch et al., 2011) with TraCI (Wegener et al., 2008) for the interface. The roundabout is connected to approach roads with give-way lines at the junctions. Besides, there is a suggested speed limit that can be exceeded by the ego vehicle as well as other vehicles. The simulation parameters for the selected roundabout and the vehicle properties are presented in Table 5 below.

| Lane width | 3.5m |
|---|---|
| Velocity limit | 13.89m/s |
| Vehicle Length | 5m |
| Max Acceleration/Deceleration | $2.6m/s^2$ |
| Max Deceleration | $4.5m/s^2$ |
| Sensor range | 42m |

Table 1: Simulation scenario and Vehicle parameters

The agent is to drive into the roundabout from the 1st entrance and drive out in the 4th exit as fast as possible. If there is no collision with other vehicles, it succeeds. The ego vehicle follows the policies computed the speed using the reinforcement learning algorithm, whereas other participants use the default car following model (Kraus) integrated in SUMO. Basically, the agent is to learn a speed profile under the default lateral control strategy of SUMO.

Besides, to show the general applicability of our methods, we designed two scenes with different density of traffic, namely the congested traffic and regular traffic. The illustration of traffic density is shown below in Figure 4. For the congested traffic, there are 12 equally distributed vehicles every 10 seconds from different entrances driving into the roundabout, while for the normal traffic, there are 12 equally spaced vehicles driving in every 30 seconds.

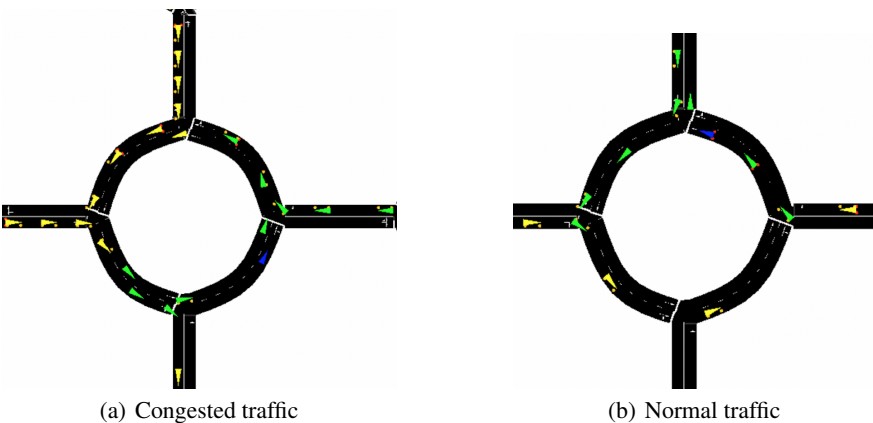

(a) Congested traffic      (b) Normal traffic

Figure 4: The figure depicts the bird's-eye view of the double-lane four-exit roundabout with an intersection network in two different traffic situations.. The number denotes the index of different exits/entrances. The ego vehicle is in blue, while other vehicles are green and yellow, indicating if vehicles are in the range of the radar. **Congested traffic**: around 24-30 vehicles are in the roundabout at the same time. **Regular traffic**: around 12-15 vehicles are in the roundabout at the same time.

## 6 RESULTS

In our experiments, we aim to answer the following questions:

- Does our method successfully learn the latent constraints?

- How does the trained agent perform compared with those guided by manually designed constraints?
- Is our method sensitive to hyperparameters?

In this part, we evaluate the convergence of the CRM and combine it with a CMDP RL algorithm safe SAC (SSAC) (Chow et al., 2019). SSAC is a safety constrained variant of the original algorithm Haarnoja et al. (2018) through Lagrangian relaxation procedure. Details of the Lagrangian-based safe baselines are referred to Appendix C. To make fair comparison, we also designed a very conservative cost with optimized structure and parameter, and also use SSAC to train the agents. It is worth mentioning that, for image-based tasks, it is generally difficult to design a cost, while our framework and algorithm is still applicable, which we leave for future work.

## 6.1 CONVERGENCE

As demonstrated in Figure 5, CRM could efficiently model the latent constraint. Besides, throughout the training, it converges with low variance even though both the model and environment are randomly initialized.

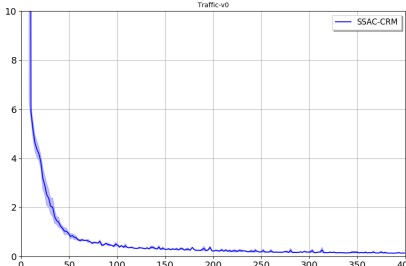

Figure 5: Training of CRM. X-axis indicates the training steps in thousand and Y-axis indicates the loss of CRM. The shadowed region shows the 1-SD confidence interval over 5 random seeds.

## 6.2 COMPARISON WITH BASELINE

In this part, we compare the performance between agents trained by SSAC with and without CRM. Specifically, we use success rate, i.e. the probability of leaving from the right exit without collision, as the metric of safety performance. As demonstrated in Figure 6, in both congested traffic and regular traffic scenarios, SSAC with CRM achieves better success rate and convergence speed than SSAC baseline.

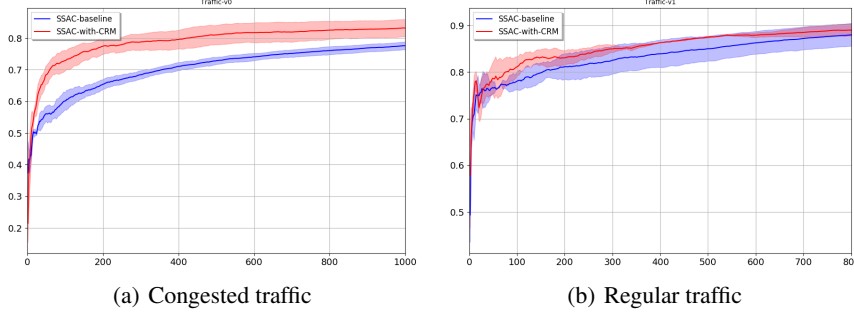

(a) Congested traffic        (b) Regular traffic

Figure 6: Success rate of agents trained by SSAC with and without CRM during training. X-axis indicates the training steps in thousand and Y-axis indicates the success rate. The shadowed region shows the 1-SD confidence interval over 5 random seeds.

### 6.3 INSENSITIVITY TO HYPERPARAMETERS

The threshold $\overline{d} \in [0, WD(p_\theta(z|s=0), p_\theta(z|s=1))]$ in 12 is a hyperparameter to be tuned. We show in this part that the performance of our approach is insensitive to this hyperparameter. We tested 3 different threshold setting $[\frac{ub}{5}, \frac{2ub}{5}, \frac{3ub}{5}]$, where $ub = WD(p_\theta(z|s=0), p_\theta(z|s=1))$. As shown in Figure 7, CRM performs stably across different settings. The Wasserstein cost is highly generalized and insensitive to hyperparameters, which is easily for implementation. To the contrary, the manually designing safety cost for complicated tasks like autonomous driving requires lots of tuning and could be very sensitive to parameters.

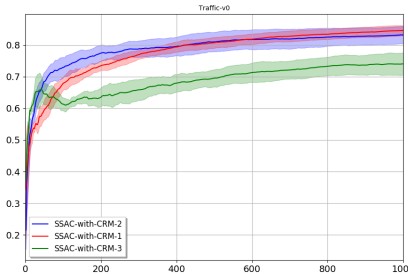

Figure 7: Success rate of agents trained by SSAC with CRM under different $\overline{d}$. X-axis indicates the training steps in thousand and Y-axis indicates the success rate. The shadowed region shows the 1-SD confidence interval over 5 random seeds.

## 7 CONCLUSION

Inspired by the situation awareness of human intelligence, in this paper, we proposed a universal approach of and latent constraints representation, which could be inherently embedded into CMDP reinforcement learning framework. We believe better situation awareness is an essential steps towards creating more intelligent agents. The roundabout experiment shows that our approach can indeed help agent aware the potential danger and improve the performance. Besides, the CRM is able to generalize through different traffic flow. Our works represent an initial step in combining RL and situation modeling. Future work includes: i) optimizing the prior distribution; ii) extending to image-based task which still a open problem for CMDP; iii) exploring other kinds of situation awareness for reinforcement learning.

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

## A  EXPERIMENT SETTING

### A.1  STATE AND ACTION

we assume the the ego vehicle could detect and track the a maximum of three other vehicles' position, orientation, and velocity. The parameters of the sensor field used are given in the Table 5. Besides, the ego vehicle is able to estimate its condition. Using the above information, the ego-state and observation space of the ego vehicle is defined as follows:

$$S_{ego} = <x, y, p, v>$$ (13)

where $x, y, p, v$ indicate the 2-D position, orientation, velocity respectively. The coordinate transformation that aligns the vehicle position and velocity along the vehicle axis makes the vehicle state invariant to road geometry.

The state or observations of other vehicles in the range of sensor field is recorded as:

$$O_i = <x_i, y_i, p_i, v_i>$$ (14)

where $O_i$ refers to the radar information of vehicle $i$. If no vehicle is present in the range of sensor field, then the $x, y$ are set to sensor range limits, and orientation and $v$ are set to 0.

Using the above and then setting that up to three vehicles could be tracked, the state of the agent at each time step can be defined as the ego state and observation of the three closest vehicles follows:

$$x_t = <S_{ego}, O_1, O_2, O_3>$$ (15)

As mentioned before, our agent aims to learn a speed profile under the default lateral control strategy of SUMO. Thus, the action can be defined as:

$$a_t = <v>$$ (16)

where $v$ is the desired velocity of ego vehicle within the range of maximum speed.

## A.2 REWARD AND COST

The agent updates policy according to the feedback of its interaction with the environment. For our scenario, the goal is to drive safely and efficiently through a roundabout without any collision with other vehicles. Thus, the reward is defined as:

$$r_t = v \tag{17}$$

where the $v$ is the current speed of ego vehicle, while the safety cost $\hat{c}(x)$ is defined in Equation 3.

## B TRAINNING DETAILS

| Hyperparameters | Intense Roundabout | Intense Roundabout |
|---|---|---|
| Minibatch size | 256 | 256 |
| Actor learning rate | 1e-4 | 1e-4 |
| Critic learning rate | 3e-4 | 3e-4 |
| Target smoothing coefficient($\tau$) | 0.005 | 0.005 |
| Discount($\gamma$) | 0.99 | 0.99 |
| Safety $\gamma$ | 0.5 | 0.5 |
| Threshold | 0.5 | 0.5 |

Table 2: SSAC Hyperparameters

For SSAC, there are two networks: the policy network and the Q network. For the policy network, we use a fully-connected MLP with two hidden layers of 256 units, outputting the mean and standard deviations of a Gaussian distribution. For the Q network and the Lyapunov network, we use a fully-connected MLP with two hidden layers of 256 units, outputting the Q value and the Lyapunov value. All the hidden layers use Relu activation function and we adopt the same invertible squashing function technique as Soft Actor-critic (SAC) Haarnoja et al. (2018) to the output layer of the policy network.

| Hyperparameters | Conditional representation model |
|---|---|
| Minibatch size | 512 |
| learning rate | 0.001 |

Table 3: CRM Hyperparameters

For CRM, there are two networks: the decoder and the encoder. For the encoder, we use a fully-connected MLP with two hidden layers of 256 and 128 units repectively, outputting the mean and standard deviations of a Gaussian distribution. For the decoder, we use a fully-connected MLP with two hidden layers of 128 and 256 units respectively, outputting the reconstruction of input. All the hidden layers use Sigmoid activation function.

The implementation of the algorithms are based on TensorFlow (Abadi et al., 2016).

## C LAGRANGIAN-BASED SAFE ALGORITHM

We include a Lagrangian based method as the baseline in our experiments, namely the safe soft actor-critic (SSAC). SSAC attempts to solve the following unconstrained optimization problem

$$\min_{\pi} \max_{\lambda \geq 0} \mathcal{L}(\pi, \lambda) = \mathbb{E}_{(s,a,s') \sim \mathcal{D}} \left[ \beta[\log(\pi_\theta(f_\theta(\epsilon, s)|s)) + \mathcal{H}_t] - Q(s, f_\theta(\epsilon, s)) + \lambda(Q_c(s, a) - \overline{d}) \right] \tag{18}$$

where $Q_c$ is the safety Q-function. By employing the same policy gradient algorithm as in SAC, a safe policy is obtained. The value of Lagrange multipliers $\beta$, $\lambda$ are adjusted by the gradient ascent method with the following objectives, respectively,

$$J(\beta) = \beta \mathbb{E}_{(s,a) \sim \mathcal{D}}[\log(\pi_\theta(a|s)) + \mathcal{H}_t] \tag{19}$$

$$J(\lambda) = \lambda \mathbb{E}_{(s,a) \sim \mathcal{D}}[L_c(s', f_\theta(\epsilon, s')) - L_c(s, a) + \alpha_3 c(s, a)] \tag{20}$$

