# OpenReview forum: "Variational Constrained Reinforcement Learning with Application to Planning at Roundabout"
_ICLR.cc/2020/Conference — Reject_

### Official Review · AnonReviewer1 · 2019-10-22
**Official Blind Review #1**

**Rating:** 1

**Review:**

The paper presented a CRM model which is VAE with separate priors for save and unsafe modes and utilize it in  RL  for roundabout planning task.

Pros:
1. The motivation of the work is clear and solves an important task
2. The approach is sensible

Cons:
1. Experimental evaluation is very weak. It is only performed on one task which is on getting out of the roundabout.
It also does not compare with any real-baseline methods. It is only the proposed method compare with the weaker version of the method.

2. Using a mixture of Gaussian for VAE has been used in many works before, such as Nazabal, Alfredo, Pablo M. Olmos, Zoubin Ghahramani, and Isabel Valera. "Handling incomplete heterogeneous data using vaes." arXiv preprint arXiv:1807.03653 (2018).
Also, make such prior should be equivalent to regular Gaussian prior with multi-head decoder as in figure1 (b) in VCL paper https://arxiv.org/pdf/1710.10628.pdf where s is T there.

So, this work did not discuss these approaches and the novelty of the work is also limited.

3. How to set the mean of s may be critical for the performance. Analysis needed.

4. The method should at least compare with conditional VAE used in the same way in RL. In this case,  the label is s and I believe that the latent space will be meaningful regarding s.

5.. The second term in the right-hand-side of equation (7) has missed some constant factor.

5. How figure 5 shows CRM could efficiently model the latent constraint? It only shows that the model converged.

6. Writing needs to be improved. There are grammatical mistakes here and there.





**Experience Assessment:**

I have read many papers in this area.

**Review Assessment: Checking Correctness Of Derivations And Theory:**

I assessed the sensibility of the derivations and theory.

**Review Assessment: Checking Correctness Of Experiments:**

I assessed the sensibility of the experiments.

**Review Assessment: Thoroughness In Paper Reading:**

I made a quick assessment of this paper.

---

### Official Review · AnonReviewer2 · 2019-10-24
**Official Blind Review #2**

**Rating:** 1

**Review:**

This paper proposes a reinforcement learning model to solve a task relevant to autonomous driving, and evaluates that model in a traffic simulator called Simulation of Urban Mobility (SUMO).

The paper describes related work but the connection to the exact problem they are solving wasn’t 100% clear to me.

The description of the model was somewhat confusing to me, but my understanding is that the model does the following:
- The dataset contains states, and each state has a label that says whether it is safe or unsafe
- However, we don’t know the implicit constraints that determine whether a state is safe or unsafe
- We learn a latent representation of the states, where we want the safe states and unsafe states to be separated in the latent space
- We use a Wasserstein distance metric in the latent space to construct a safety cost function

The paper gives pseudocode for the algorithm, and experiments in a traffic simulator of the task of driving through a two-lane roundabout with four exits.

I don’t see a significant enough algorithmic contribution from this paper to yield an ICLR acceptance. I think this paper would be better suited for a more application-specific conference that would be more interested in the empirical progress on a task specific to autonomous vehicles.

As a last note, the paper contains a large number of grammatical errors and typos - I’m not considering these in the score, but the paper would benefit from a close proofreading by a native English speaker.

**Experience Assessment:**

I do not know much about this area.

**Review Assessment: Checking Correctness Of Derivations And Theory:**

N/A

**Review Assessment: Checking Correctness Of Experiments:**

I did not assess the experiments.

**Review Assessment: Thoroughness In Paper Reading:**

I read the paper at least twice and used my best judgement in assessing the paper.

---

### Official Review · AnonReviewer4 · 2019-10-28
**Official Blind Review #4**

**Rating:** 1

**Review:**

This paper tackles the challenge of control with safety constraints using a learned soft constraint formulation. It models data as coming from a hierarchical generative model p(s) p(z|s) p(x|z,s) where s is a safety indicator variable, z is a latent variable, and x is an observation. Then it learns a variational approximation q(z|x) to the true posterior p(z|x). By then measuring a divergence between q(z|x) and p(z|s), the authors aim to evaluate whether a state is safe or unsafe. They then combine this learned safety constraint with a constrained RL method, SSAC, to learn safe policies.

While this setting is interesting and worthy of further exploration, there are sufficient issues with this work that it should not be accepted at this time.

For one thing, the motivation behind the constrained MDP formulation of reinforcement learning, such as the original SSAC paper, is to provide safety throughout training. However, this work learns a soft constraint over the course of training by using examples where the policy leads to catastrophic failures. This means that, unlike SSAC, this work does not encourage safety at all early in training. Since it collects a large number of catastrophic experiences, this method is not categorically different from e.g. training a policy with large negative rewards for crashing.

Furthermore, since there is no constraint preventing crashing, a poorly-performing learned constraint might lead to better performance as measured by reward alone. Since safety during training comes with a tradeoff against rewards, as shown by SSAC, a poorly-functioning learned constraint might lead to improved reward. This work lacks an evaluation of the number of crashes to complement the rewards depicted in Figure 6.

The particulars of the method used for computing whether the safety constraint is violated are somewhat surprising. The authors use a Wasserstein distance to compute a cost as a function of the q(z|x) and p(z|s=1). They motivate this choice by the fact that the KL divergence would go to infinity for non-overlapping distributions; however, in equation (12) that would not significantly affect the computed cost. Since this divergence is calculated in the low-dimensional latent space and one of the distributions is fixed, it is also unclear that this would ever arise. It also seems that a likelihood ratio test between p(x|s=1) and p(x|s=0) would provide a more meaningful signal than simply classifying whether a point is near the "dangerous" prior p(z|s=1).

Overall I think there are some interesting ideas inside this work, but it needs some improvements:
1. Reframing to make the setting make more sense. If you want to compare against SSAC, you need to be minimizing the total number of catastrophic events during training. It might make sense to assume a pre-existing set of "dangerous" examples, e.g. labeled ahead of time by a human.
2. Textual editing and work to make the notation more consistent, e.g. the top of page 3 uses s for states as well as the "safe" indicator.
3. Significantly improved evaluation. The results here lack crucial information about the number of catastrophic events during training. I would also like to see ablations or probe experiments showing what is learned by the encoder. Furthermore, this one environment is extremely simple (one degree of freedom) and to have confidence that the method works more generally, I would like to see the method applied to richer tasks.

**Experience Assessment:**

I have published one or two papers in this area.

**Review Assessment: Checking Correctness Of Derivations And Theory:**

I assessed the sensibility of the derivations and theory.

**Review Assessment: Checking Correctness Of Experiments:**

I assessed the sensibility of the experiments.

**Review Assessment: Thoroughness In Paper Reading:**

I read the paper at least twice and used my best judgement in assessing the paper.

---

### Decision · Program_Chairs · 2019-12-19

**Decision:**

Reject

**Comment:**

This paper proposes to add constraints to the RL problem within a variational method. The hope is to specify a safe vs non-safe states. The reviewers were not convinced that this paper makes the cut for ICLR. Moreover, there was no rebuttal from the authors, so it didn't give the reviewer a chance to reconsider their opinion. Based on the current ratings, I recommend to reject this paper.